# In Situ Construction of a MgSn(OH)_6_ Perovskite/SnO_2_ Type-II Heterojunction: A Highly Efficient Photocatalyst towards Photodegradation of Tetracycline

**DOI:** 10.3390/nano10010053

**Published:** 2019-12-24

**Authors:** Yuanyuan Li, Xiaofang Tian, Yaoqiong Wang, Qimei Yang, Yue Diao, Bin Zhang, Dingfeng Yang

**Affiliations:** 1Department of Biological and Chemical Engineering, Cooperative Innovation Center of Lipid Resources and Children’s Daily Chemicals, Chongqing University of Education, Chongqing 400067, China; tianxiaofang0717@163.com (X.T.); yang53075@163.com (Q.Y.); yuediao704@163.com (Y.D.); 2College of Chemistry and Chemical Engineering, Chongqing University of Technology, 69 Hongguang Rd., Lijiatuo, Banan District, Chongqing 400054, China; wangyq863@cqut.edu.cn; 3Analytical and Testing Center of Chongqing University, Chongqing 401331, China

**Keywords:** perovskite-type hydroxide, photocatalysis, tetracycline, photoelectrochemistry, type-II heterojunction

## Abstract

Using solar energy to remove antibiotics from aqueous environments via photocatalysis is highly desirable. In this work, a novel type-II heterojunction photocatalyst, MgSn(OH)_6_/SnO_2,_ was successfully prepared via a facile one-pot in situ hydrothermal method at 220 °C for 24 h. The obtained heterojunctions were characterized via powder X-ray diffraction, Fourier-transform infrared spectroscopy, transmission electron microscopy, and ultraviolet-visible diffuse reflectance spectroscopy. The photocatalytic performance was evaluated for photodegradation of tetracycline solution under ultraviolet irradiation. The initial concentration of tetracycline solution was set to be 20 mg/L. The prepared heterojunctions exhibited superior photocatalytic activity compared with the parent MgSn(OH)_6_ and SnO_2_ compounds. Among them, the obtained MgSn(OH)_6_/SnO_2_ heterojunction with MgCl_2_·6H_2_O:SnCl_4_·5H_2_O = 4:5.2 (mmol) displayed the highest photocatalytic performance and the photodegradation efficiency conversion of 91% could be reached after 60 min under ultraviolet irradiation. The prepared heterojunction maintained its performance after four successive cycles of use. Active species trapping experiments demonstrated that holes were the dominant active species. Hydroxyl radicals and superoxide ions had minor effects on the photocatalytic oxidation of tetracycline. Photoelectrochemical measurements were used to investigate the photocatalytic mechanism. The enhancement of photocatalytic activity could be assigned to the formation of a type-II junction photocatalytic system, which was beneficial for efficient transfer and separation of photogenerated electrons and holes. This research provides an in situ growth strategy for the design of highly efficient photocatalysts for environmental restoration.

## 1. Introduction

Pharmaceutical residues in wastewater are a severe threat to the ecological environment. Misuse and overuse of antibiotics can lead to high antibiotic resistance and multi-resistant strains of microorganisms [1,2,3,4]. As a broad-spectrum antibiotic, tetracycline (TC) is widely used for disease treatment in humans, and it is frequently detected in various bodies of water owing to its high solubility. Removal of TC from wastewater is important. Semiconductor-based photocatalysis, an advanced oxidation method, is the most promising technique for harvesting solar energy to alleviate environmental problems [5,6]. The discovery and design of photocatalysts that are stable, efficient, inexpensive, and ecofriendly is of importance [7,8]. Among the numerous photocatalytic materials, perovskite-based materials have excellent stability, crystalline structures with high symmetry, and diverse chemical and physical properties [9,10].

Recently, a perovskite-type hydroxide, MSn(OH)_6_ (M = Mg, Mn, Fe, Co, Ni, Cu, Zn, Sr, or Ba), was prepared via the hydrothermal method and showed excellent potential as a photocatalyst for environmental remediation [11,12]. In the crystalline structure of this compound, the metal ions are octahedrally coordinated with OH to form M(OH)_6_ and Sn(OH)_6_ polyhedra that connect with each other by sharing O atoms to build the whole crystalline structure. Because they have different Shannon radii, these hydroxides exhibit multiple crystalline symmetries. Compounds containing Mg, Ca, Mn, Fe, Co, Ni, Cu, Zn, or Cd adopt the rock salt ReO_3_ structure with cubic or tetragonal symmetry [13,14,15,16]. In contrast, those containing Sr or Ba exhibit a distorted perovskite structure in the monoclinic family with the space group *P*2_1_/n [14]. The wide variety of compositions and constituent elements in these hydroxide perovskites offers plenty of room to study their photocatalytic performance. Considerable effort has been devoted to study their applications in environmental and energy catalysis research. Nanocubes of CaSn(OH)_6_ and CuSn(OH)_6_ with a d^10^-d^10^ electronic configuration exhibit high degradation rates for methylene blue [17,18,19]. The CoSn(OH)_6_ catalyst has high activity and selectivity for the photoreduction of CO_2_ to CO [20]. MgSn(OH)_6_ and ZnSn(OH)_6_ could be applied to degradation of gaseous C_6_H_6_ and reforming of ethanol to H_2_ and CH_4_ [11,21]. These studies have shown that hydroxide perovskites have excellent potential for photocatalysis. However, few studies have investigated the ability of this family of compounds to remove antibiotics in photocatalysis.

Generally, single photocatalysts have some drawbacks, such as the rapid recombination of photogenerated electrons and holes and relatively small reactive sites, which will limit their practical application [22]. Construction of heterostructures is an effective method to overcome these shortcomings and numerous reports have demonstrated that the interface formed between two different semiconductors could accelerate carrier transfer, and the recombination rate of the spatially separated photogenerated carriers can be reduced, resulting in greatly improved photocatalytic performance [23,24,25,26]. A high-quality heterojunction with a good lattice match is highly desired, and could be fabricated by co-sharing of one or several of the same atoms between the formed coherent or semi-coherent interface.

In this work, SnO_2_ was investigated for coupling with MgSn(OH)_6_ because of its chemical stability, low exciton energy, and outstanding conductivity. The simulated electron and hole mobilities along the *x*-direction were reported to be ~2966 and 66 cm^2^/Vs, respectively. Because of the large difference in transport behavior, SnO_2_ has been widely used as an efficient photocatalyst [27]. Furthermore, at the atomic level, an intimately contacted interface by co-sharing of the Sn atoms would be expected and can be fabricated via a one-pot in situ hydrothermal treatment between MgSn(OH)_6_ and SnO_2_. The observed SnO_2_ nanoparticles were loaded tightly on the surface of the MgSn(OH)_6_ nanocubes. As a result, photoinduced carriers in the heterojunction were effectively separated and the photocatalytic activity was remarkably improved for degradation of a TC solution under ultraviolet (UV) irradiation. Finally, a possible photocatalytic mechanism for the type-II heterojunction is proposed and discussed.

## 2. Materials and Methods

### 2.1. Synthesis of SnO_2_, MgSn(OH)_6_, and MgSn(OH)_6_/SnO_2_ Heterojunctions

The following materials were applied for this experiment: MgCl_2_·6H_2_O (Aladdin, 98%, Shanghai, China), SnCl_4_·5H_2_O (Aladdin, 99%, Shanghai, China), tetracycline (Adamas-beta, 97%+, Shanghai, China), NaOH (Aladdin, 96%, Shanghai, China), ethyl alcohol (Aladdin, 99.7%, Shanghai, China), 2-propanol (IPA) (Aladdin, 99.7%, Shanghai, China), disodium ethylenediaminetetraacetic acid (EDTA) (Aladdin, 99.5%, Shanghai, China), and 2,2,6,6-tetramethylpiperidine-1-oxyl (TEMPO) (Aladdin, 98%, Shanghai, China). All of the above chemicals were directly utilized without any further purification.

To obtain pure phase SnO_2 _(SO), 4 mmol SnCl_4_·5H_2_O was dissolved in 15 mL of water. Then, 4 mL NaOH (0.5 g/mL) was added to adjust pH to ~7 and the mixture was further stirred for 5 min. Next, the mixture was transferred into a 25-mL Teflon-lined steel autoclave and hydrothermally heated at 220 °C for 24 h.

For preparation of pure MgSn(OH)_6_ (MSOH), 4 mmol MgCl_2_·6H_2_O, 4 mmol SnCl_4_·5H_2_O, and 12 mL NaOH (0.5 g/mL) were added to 15 mL of deionized water to adjust pH to ~11 and magnetically stirred for further 10 min. The obtained slurry was hydrothermally treated at 220 °C for 24 h in a Teflon-lined steel autoclave.

Heterojunctions of MgSn(OH)_6_/SnO_2_ with different mass fractions were prepared by an in situ hydrothermal formation strategy. The mass fraction of SnO_2_/MgSn(OH)_6_ was adjusted to 10%, 20%, 30%, or 40% by controlling the amount of SnCl_4_·5H_2_O added, and the obtained products were labeled as MSOH-SO-1 (SnCl_4_·5H_2_O/MgCl_2_·6H_2_O = 4.4/4 mmol), MSOH-SO-2 (SnCl_4_·5H_2_O/MgCl_2_·6H_2_O = 4.8/4 mmol), MSOH-SO-3 (SnCl_4_·5H_2_O/MgCl_2_·6H_2_O = 5.2/4 mmol), and MSOH-SO-4 (SnCl_4_·5H_2_O/MgCl_2_·6H_2_O = 5.6/4 mmol), respectively. Taking MSOH-SO-3 as an example, 4 mmol MgCl_2_·6H_2_O and 5.2 mmol SnCl_4_·5H_2_O were dispersed in 15 mL of deionized water, followed by addition of were 4 mL NaOH (0.5 g/mL) to adjust pH to ~8. The mixture was further magnetically stirred at room temperature for 30 min and then put into a 25-mL Teflon-lined steel autoclave and kept in an oven at 220 °C for 24 h.

All the products were washed with distilled water several times and then dried at 70 °C for 10 h.

### 2.2. Characterization

Powder X-ray diffraction (PXRD) was performed on a PANalytical X’pert powder diffractometer equipped with a PIXcel detector and using Cu Kα radiation (40 kV and 40 mA). A scan step width of 0.01° and a scan rate of 0.1° s^−1^ were applied to record patterns in the 2θ range of 6°–90°. Transmission electron microscopy (TEM), high-angle annular dark field imaging (HAADF), and energy-dispersive X-ray spectrometry of the MgSn(OH)_6_/SnO_2_ heterojunctions were performed using a Talos F200S G2 microscope to investigate the microstructures of the samples. Ultraviolet-Visible diffuse reflectance spectroscopy (UV–Vis DRS) data were collected at room temperature using a powder sample with BaSO_4_ as a standard on a Shimadzu UV-3150 spectrophotometer over the spectral range of 200–800 nm. Fourier transform infrared (FT-IR) spectra were collected using a Nicolet 360 spectrometer with a 2 cm^–1^ resolution in the range of 500–4000 cm^−1^. The photocatalysts were fixed within a pressed KBr pellet. For instance, 1 mg heterojunction MSOH-SO-3 within 100 mg KBr were pressed at 15 MPa for 15 min, forming 13 mm pellets. Total dissolved organic carbon (TOC) was determined via a TOC analyzer (SHIMADZU, TOC-L CPB). The BET specific surface areas were investigated by means of N_2_ adsorption-desorption at 77 K using a Quantachrome QuadraWin and the specific surface areas were determined according to the BET method in the relative pressure range p/p^0^ = 0.069~0.249. Electrochemical measurements were conducted on a CHI 660E workstation. A Pt plate, calomel electrode, and MSOH-SO-3 coated on indium tin oxide (ITO) were used as the counter electrode, reference electrode, and working electrode, respectively, in a three-electrode cell. For a typical preparation of working electrode, 10 mg of catalyst samples were dispersed in 500 μL of water/ethanol (240/250) mixed solvent containing 10 μL of 5 wt % Nafion and sonicated for 30 min. Then, a certain amount of the catalyst ink was loaded onto an ITO electrode. Electrochemical impedance spectroscopy (EIS) was carried out using an alternating voltage with an amplitude of 5 mV over the frequency range of 10^5^ Hz to 0.1 Hz and an open circuit voltage in 0.5 M Na_2_SO_4_. For analysis of the transient photocurrent responses, a 175 W high-pressure Hg lamp and Na_2_SO_4_ were used as the light source and electrolyte, respectively. Mott–Schottky curves were measured in a Na_2_SO_4_ solution with an amplitude perturbation of 5 mV and frequency of 1000 Hz.

### 2.3. Photocatalytic Activity Measurements

The photocatalytic activities of the products were evaluated for degradation of a TC solution. The light source was an external 175 W high-pressure mercury lamp with a primary wavelength of 365 nm. The lamp was positioned 9 cm above the reaction vessel and the light intensity on the surface of the suspension was approximately 11 mW/cm^2^. The reactant solution was maintained at room temperature using a stream of cool water during the photocatalytic reaction. Before irradiation, the photocatalyst powder (30 mg) and TC solution (20 mg L^−1^, 100 mL) were fully stirred in the dark for 1 h to establish the adsorption–desorption equilibrium and the pH was measured to be ~5.5. Then, the solution was exposed to the light. Aliquots (5 mL) of the suspension was taken at set intervals (20 min) and separated by centrifugation. Then, the concentration of the TC solution was determined by UV–Vis spectrometry at 355 nm and the process of photocatalytic performance was carried out within 60 min. Considering the small loss of the catalysts in the recycling process, several batches of repeated experiments for each cycle were performed. Then, the catalysts were collected and mixed to maintain the weight of 30 mg for each test. The degradation rate could be calculated by the formula
Photodegadation (%) = 1− C/C_0_(1)
where C_0_ was the absorbance of the initial solution and C was the absorbance of solution at a given time after the photocatalytic reaction. Trapping experiments of the active species were carried out using 30 mg of MSOH-SO-3 and 100 mL of TC solution (20 mg/L). The reactive intermediate participating in the degradation process was identified by using different sacrificial agents. Such as, 10 mL of 2-propanol (IPA), 0.1 mmol disodium ethylenediaminetetraacetic acid (EDTA), and 0.1 mmol 2,2,6,6-tetramethylpiperidine-1-oxyl (TEMPO) were added sequentially to trap ·OH, *h*^+^, and ·O_2_^−^ radicals, respectively. TOC analysis was carried out at every 20 min to check the degree of mineralization over heterojunction MSOH-SO-3 under UV light irradiation.

## 3. Results and Discussion

Photocatalysts synthesized from SnO_2_, MgSn(OH)_6_, and the MSOH-SO heterojunction were characterized by PXRD (Figure 1). Peaks at 19.7°, 22.8°, 32.5°, and 52.6° matched well with the crystalline planes (111), (200), (220), and (420), respectively, that belong to the cubic phase MgSn(OH)_6_ (JCPDS 074-0366). Major diffraction peaks at 26.7° (110 plane), 33.8° (101 plane), and 51.7° (211 plane) were indexed to the SnO_2_ phase (JCPDS 021-1250). Notably, the peak intensity of SnO_2_ gradually strengthened with increasing SnCl_4_·5H_2_O concentration, indicating that the amount of SnO_2_ loaded on MgSn(OH)_6_ increased. The structures of these photocatalysts were confirmed by FT-IR spectroscopy. Specifically, for SnO_2_, a broad peak located at ~618 cm^−1^ was attributed to the antisymmetric vibrational mode of Sn-O-Sn [28]. For MgSn(OH)_6_, absorption peaks in the range 3000–3500 cm^−1^ were ascribed to O–H stretching vibration modes and lattice modes at 1175 cm^−1^ corresponded to the O–H bending vibrations. Peaks in the range of 500–1000 cm^−1^ were attributed to the Mg–O and Sn–O bond stretching vibration modes [29]. All of the main characteristic peaks of MgSn(OH)_6_ and SnO_2_ were observed in MSOH-SO. On the basis of the PXRD and FT-IR observations, we concluded that both SnO_2_ and MgSn(OH)_6_ existed in the heterojunctions.

The typical interface structure of the synthesized heterojunction MSOH-OH-3 was further characterized by HAADF–TEM observation. The MgSn(OH)_6_ nanocubes were combined with randomly oriented nanoparticles of SnO_2_ (Figure 2a). Furthermore, well-ordered lattice fringes with a lattice spacing 0.38 nm could be ascribed to the (200) plane of MgSn(OH)_6_. The interplanar distances of 0.32 nm and 0.25 nm could be assigned to the (110) and (101) planes of the SnO_2_ nanoparticles, respectively. Elemental mapping via HAADF–energy-dispersive X-ray spectroscopy (Figure 2b) confirmed the uniform distributions of Mg, Sn, and O elements in the MSOH-SO-3 heterojunction. The contents of Sn and O elements were much higher than that of Mg, which showed that a heterojunction formed between MgSn(OH)_6_ and SnO_2_ (Appendix A). All of the above observations demonstrated that there was close contact at the interface of the synthesized heterojunction, which might be favorable for promoting the separation of photogenerated carriers at the interface during the photocatalytic reaction. Besides, we also measured the BET specific surface areas of the as-synthesized samples, heterojunction MSOH-SO-3, SO, and MSOH. As presented in Appendix A, the results showed that the calculated specific surface areas of sample MSOH-SO-3 and SO were 114.2 m^2^/g and 81.7 m^2^/g, respectively. However, the obtained specific surface area of MSOH was too small to ignore.

To investigate the optical absorption properties and forbidden band gap of the synthesized photocatalysts, the UV–Vis DRS spectra of MgSn(OH)_6_, MSOH-SO-3, and SnO_2_ were recorded (Figure 3). All the photocatalysts showed an obvious absorption edge in the UV region. The absorption edges for MgSn(OH)_6_ and SnO_2_ were at approximately 303 nm and 343 nm, respectively. The corresponding band gaps estimated from the curve of ${\left(\alpha h\nu \right)}^{2}$ versus the photon energy (hν) were 4.09 eV (Figure 3b) and 3.61 eV (Figure 3d), respectively. The MSOH-SO-3 heterojunction exhibited a blue shift compared with SnO_2 _(Figure 3c), which would facilitate the separation of photoinduced carriers and improve the photocatalytic performance.

The photocatalytic activities of the prepared catalysts were evaluated for photodegradation of a TC solution under UV irradiation (Figure 4). Blank experiments showed that UV light had little effect on photodegradation of the TC solution (Figure 4a). The degradation rates of pure SnO_2_ and MgSn(OH)_6_ were 44% and 34%, respectively. However, when SnO_2_ was combined with MgSn(OH)_6_ to form a heterojunction, the photocatalytic activity was enhanced. Among the heterojunctions, MSOH-SO-3 exhibited the best photocatalytic ability and the photodegradation rate reached nearly 91% in 60 min, which was much larger than that of state-of-art UV light responsive ZnO [30]. The corresponding changes in the characteristic absorption of the TC solution are shown in Appendix A. The behavior of the photocatalytic reaction was described well by a pseudo-first order model [31,32,33]
(2)$$-\ln\frac{C}{C_{0}}=kt,$$
where $c_{0}$ and c are the TC concentrations in solution at time 0 and *t*, respectively; and k is the fitted kinetic rate constant. A linear relationship was observed between $\ln\frac{C_{0}}{C}$ and *t* (Figure 4b), which revealed that the synthesized catalysts for photodegradation of TC obeyed the model perfectly. The fitted rate constants of MgSn(OH)_6_ and SnO_2_ were 0.006 min^−1^ and 0.009 min^−1^, respectively (Figure 4c). The *k* values of the heterojunctions were all higher than those of the parent compounds, and the corresponding rate constants of MSOH-SO-1, MSOH-SO-2, MSOH-3, and MSOH-SO-4 were 0.018 min^−1^, 0.024 min^−1^, 0.030 min^−1^, and 0.028 min^−1^, respectively. Among the heterojunctions, MSOH-SO-3 had the highest *k*, which was approximately five times that of MgSn(OH)_6_ and 3.3 times that of SnO_2_. The photocatalytic activity was further evidence of successful preparation of the heterojunctions and an efficient synergistic effect at the interface. The total organic carbon (TOC) performance was also performed to check mineralization ability over heterojunction MSOH-SO-3. As indicated in Appendix A, after 60 min irradiation, the TOC removal efficiency of mineralizing TC molecule could reach 35%.

To evaluate its practicality as a photocatalyst, the stability of MSOH-SO-3 was assessed in four successive cycles of reuse in photocatalytic experiments. The photocatalytic performance showed no obvious loss after four cycles (Figure 4d) and the observed PXRD patterns remained the same (Appendix A). Statistically, the standard deviations of kinetic rate constants of the four cycling performance is 0.00299. These results show that the MSOH-SO-3 heterojunction is a stable and reusable photocatalyst for the degradation of TC under UV irradiation.

The photocurrent response and EIS showed a close association between the migration, transfer, and recombination of photoinduced electron–hole pairs in the photocatalysts. The photocurrent intensities of MgSn(OH)_6_, SnO_2_, and MSOH-SO-3 were measured for four cycles of intermittent (on–off) UV irradiation (Figure 5a). The photocurrent intensities of MgSn(OH)_6_ were lower than 0.4 µA/cm^2^, whereas the photocurrent intensities of SnO_2_ were lower than 0.7 µA/cm^2^. After the coupling of SnO_2_ with MgSn(OH)_6_, a clear enhancement of the photocurrent response was observed, which suggested that the electric field at the interface of the heterojunction could accelerate the movement of carriers and greatly promote separation of the photoinduced electrons and holes. Additionally, EIS was conducted to understand the carrier transfer resistance using the arc radius [34]. Nyquist plots with smaller arc radii indicate a low impedance and a high carrier transfer efficiency. MSOH-SO-3 had the smallest arc radius compared with the parent compounds MgSn(OH)_6_ and SnO_2_ (Figure 5b). The low electron transfer resistance of MSOH-SO-3 confirmed successful fabrication of the MgSn(OH)_6_/SnO_2_ heterojunction, which could effectively improve the efficiency of electron–hole pair separation and increase the photocurrent intensity and photodegradation activity.

Trapping experiments of the reactive species during the photocatalytic process were performed to investigate the photocatalytic mechanism. For this, MSOH-SO-3. EDTA, TEMPO, and IPA were applied separately as scavengers to determine the concentrations of *h*^+^, ·O_2_^−^, and ·OH, respectively. Introduction of TEMPO and IPA in the photocatalytic process had little effect on the degradation of the TC solution (Figure 6), indicating that ·O_2_^−^ and ·OH play minor roles in the photodegradation process. In contrast, following the addition of EDTA, a dramatic inhibition of the photocatalytic efficiency was observed and the degradation rate decreased rapidly to 27%. Thus, it could be inferred that *h*^+^ was an active participant in photodegradation of TC with the MSOH-SO-3 heterojunction.

The band edge positions of MgSn(OH)_6_ and SnO_2_ were determined using the Mott–Schottky technique. Generally, the quasi Fermi level (flat band potential $V_{fb}$) can be obtained from the *x*-intercept of a Mott–Schottky plot ($\frac{1}{C^{2}}=0$) as a function of the applied potential according to the following formula [35,36]
(3)$$\frac{1}{C^{2}}=\left(\frac{2}{\varepsilon_{r}\varepsilon_{0}N_{d}e}\right)\times \left(V-V_{fb}-\frac{\kappa_{B}T}{e}\right),$$
where *C* is the space charge capacitance; $\varepsilon_{r}$ and $\varepsilon_{0}$ are the dielectric constant of the semiconductor and permittivity in vacuum, respectively; *e* is the electronic charge; $N_{d}$ is the carrier density; and V, $V_{fb}$,$\;\kappa_{B}$, and T are the applied voltage, flat-band potential, Boltzmann constant, and temperature, respectively. The flat band potential $V_{fb}$ was set to 0.1 V below the conduction band (CB) minimum or above the valence band (VB) maximum for n-type and p-type semiconductors, respectively [37]. The Mott–Schottky plots of MgSn(OH)_6_ and SnO_2_ (Figure 7a,b) had positive slopes, which indicated that they were n-type semiconductors. The estimated $V_{fb}$ of MgSn(OH)_6_ and SnO_2_ were −1.155 V and −0.655 V versus the saturated calomel electrode (SCE), respectively, and could be calibrated to normal hydrogen electrode potentials as [38,39]
(4)$$V_{RHE}=V_{SCE}+0.059pH+V_{SCE}^{0},$$
where $V_{RHE}$ is the calibrated potential versus the reversible hydrogen electrode(RHE), $V_{SCE}^{0}$ is 0.245 V, and $V_{SCE}$ is obtained from experimental values. Thus, the $V_{fb}$ of MgSn(OH)_6_ and SnO_2_ were −0.505 and −0.005 V, respectively, versus RHE after calibration. Therefore, the calibrated CB minima of MgSn(OH)_6_ and SnO_2_ were −0.605 and −0.105 V, respectively. According to the optical band gaps from the UV–Vis DRS curves, the valence band maxima of MgSn(OH)_6_ and SnO_2_ were 3.485 and 3.505 V, respectively.

In light of the above experimental results and analysis, a possible photocatalytic mechanism for TC degradation using the MSOH-SO-3 heterojunction is proposed in Figure 8. According to the calculated band position, MgSn(OH)_6_ and SnO_2_ exhibited type-II band alignment, and an energy bias was naturally generated at the contact interfaces between the two compounds. This facilitated the separation and transfer of photoinduced carriers. Under UV irradiation, both MgSn(OH)_6_ and SnO_2_ could be excited to produce electrons and holes. The excited electrons on the CB of MgSn(OH)_6_ and LOMO level of TC molecule tended to flow down to the CB of SnO_2_ across the interface. Meanwhile, the remaining holes on the VB of SnO_2_ moved to the VB of MgSn(OH)_6_ and HOMO level of TC molecule. This transfer mechanism greatly prolonged the relaxation time of the photoinduced carriers, which agreed with the photoelectrochemical measurement results. The accumulated electrons in the CB position of SnO_2_ were more positive than O_2_/·O_2_^−^, thus ·O_2_^−^ would theoretically not participate in the photocatalytic reaction. By contrast, the accumulated holes *h*^+^ in MgSn(OH)_6_ could capture OH^−^ to generate ·OH thermodynamically. However, there was no large decrease in the degradation of the TC solution in the presence of IPA according to our reactive species trapping experiments. These results indicated that the produced ·OH would not be favorable for the oxidation of TC, as reported earlier [40]. Thus, the *h*^+^ present in the VB of MgSn(OH)_6_ could directly oxidize the TC solution to the corresponding degradation products. Therefore, formation of the type-II heterojunction that accelerated the carrier separation could be considered as the primary factor for the enhanced photocatalytic activity towards the TC degradation.

## 4. Conclusions

In summary, a series of novel MSOH-SO heterojunction photocatalysts containing n-type MgSn(OH)_6_ and n-type SnO_2_ were successfully prepared via a facile one-pot in situ method. The heterojunction MSOH-SO-3 exhibited outstanding photodegradation activity for TC solution and the maximum photodegradation rate was 91% after UV irradiation within 60 min. This rate was approximately 5 times that of MgSn(OH)_6_ and 3.3 times that of SnO_2_. The enhancement of the photocatalytic performance could be attributed to the successful preparation of a type-II heterojunction between MgSn(OH)_6_ and SnO_2_, which greatly enhanced the efficient separation of photoinduced carriers based on the structural analysis, photoelectrochemical measurement and reactive species detecting experiments. The photoinduced *h*^+^ radicals played a dominant role in the photocatalytic degradation of a TC solution over the MSOH-SO-3 heterojunction photocatalyst. This research might provide a facile way for construction highly efficient heterojunction photocatalysts.

## Figures and Tables

**Figure 1 nanomaterials-10-00053-f001:**
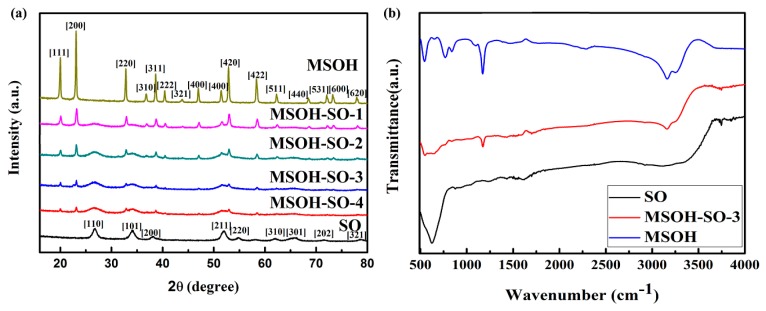
(**a**) XRD patterns with miller indices of several strongest peaks; (**b**) FT-IR spectra of SnO_2_, MgSn(OH)_6_, and MSOH-SO-3 heterojunction.

**Figure 2 nanomaterials-10-00053-f002:**
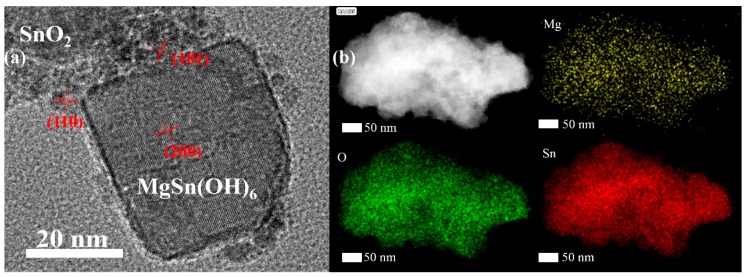
(**a**) HRTEM image; (**b**) HAADF and EDX spectrum of the MSOH-SO-3 heterojunction.

**Figure 3 nanomaterials-10-00053-f003:**
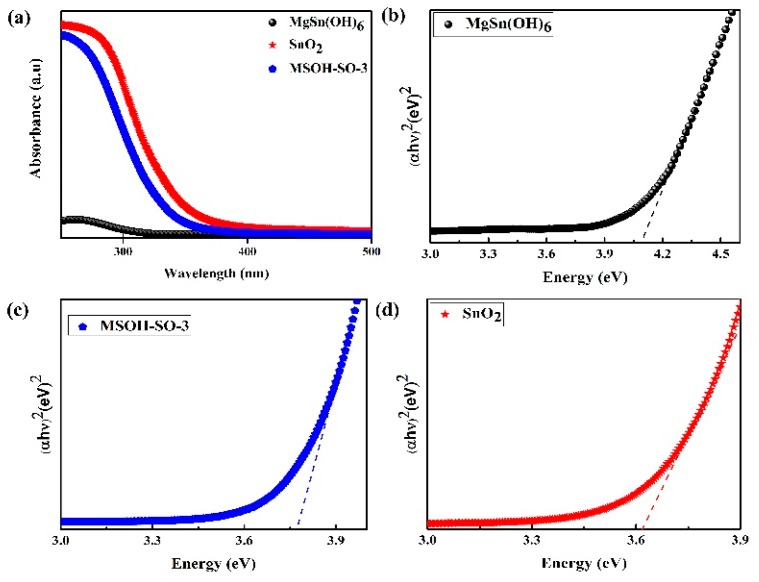
(**a**) UV–Vis DRS and plots of ${\left(\alpha h\nu \right)}^{2}$ versus the photon energy (hν) for the band gap energies of (**b**) MgSn(OH)_6_, (**c**) MSOH-SO-3, and (**d**) SnO_2_.

**Figure 4 nanomaterials-10-00053-f004:**
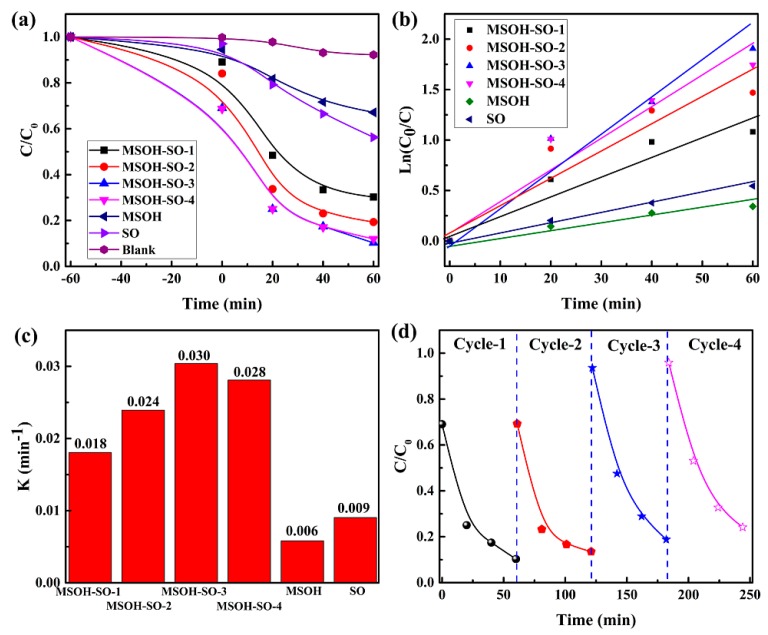
(**a**) Photodegradation of TC solution with the synthesized catalysts under UV irradiation. (**b**) Linear fitting with a pseudo-first order reaction model for TC photodegradation with the synthesized samples. (**c**) The fitted kinetic constant for photocatalytic reaction of TC solution. (**d**) Results for four cycles of reuse of MSOH-SO-3 for TC photodegradation.

**Figure 5 nanomaterials-10-00053-f005:**
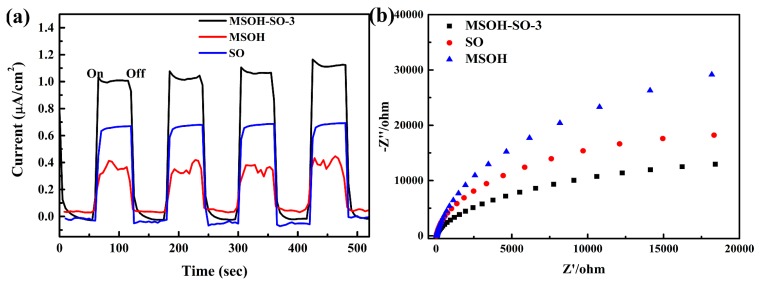
(**a**) Transient photocurrent responses under UV–Vis irradiation. (**b**) EIS Nyquist plots of MgSn(OH)_6_, SnO_2_, and MSOH-SO-3.

**Figure 6 nanomaterials-10-00053-f006:**
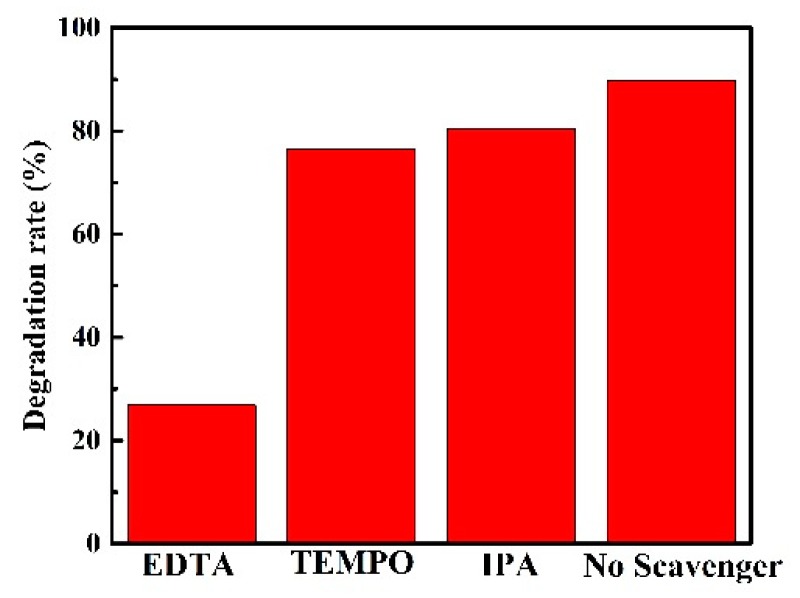
Trapping experiment for the reactive species during TC photodegradation over the MSOH-SO-3 heterojunction.

**Figure 7 nanomaterials-10-00053-f007:**
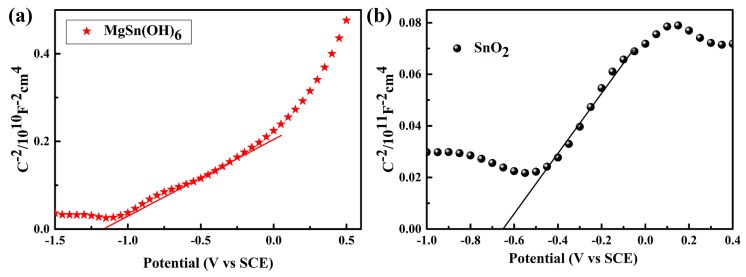
Mott–Schottky curves of (**a**) MgSn(OH)_6_ and (**b**) SnO_2._

**Figure 8 nanomaterials-10-00053-f008:**
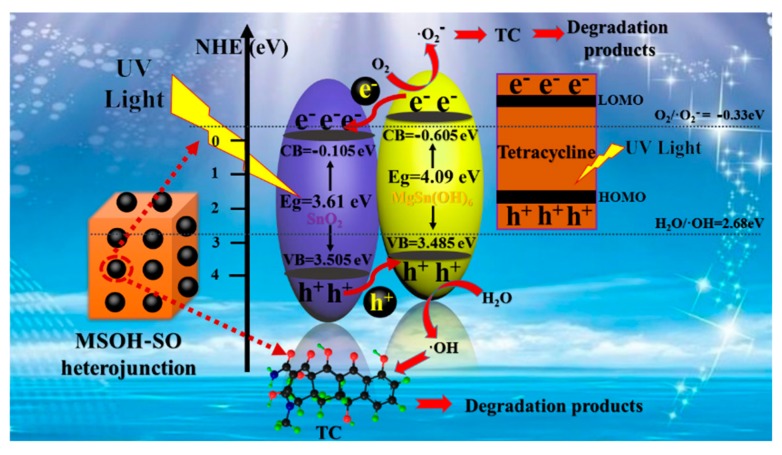
Schematic of the carrier transfer in a type-II heterojunction between MgSn(OH)_6_ and SnO_2_.

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
