# Peer review of "In Situ Construction of a MgSn(OH)6 Perovskite/SnO2 Type-II Heterojunction: A Highly Efficient Photocatalyst towards Photodegradation of Tetracycline"

_nanomaterials, 2019, doi:10.3390/nano10010053_

Round 1

Reviewer 1 Report

In this manuscript, a novel type-II heterojunction photocatalysts, MgSn(OH)6/SnO2, were successfully constructed through a facile one-pot in-situ hydrothermal method. The obtained products are thoroughly characterized by XRD, FT-IR, HRTEM, optical, and catalytic properties. The photocatalytic performance was examined for the degradation of a tetracycline solution under UV light irradiation. The manuscript is well organized and contains interesting findings. However, I recommended a major revision of the article from its present form before it can be published in nanomaterials. The main concerns are listed below.

The authors should explain the novelty of the present report in a scientific manner? The authors should provide the schematic representation for the experimental procedure. The authors should include the line spectra and miller indices in the XRD pattern. What is the interaction between MgSn(OH)6 and SnO2, which makes it stable? chemical bond? physical force? how to confirm that heterojunction really form by direct proof.? why is it not just a composite? although their band matches. How to exclude the sensitization mechanism of tetracycline for the present work? TOC is needed to sure the mineralization ratio. Photoluminescence spectra are needed to explore the mechanism. The authors should check the loading effect of the catalyst for the degradation of dye. The latest, important publications are suggested, such as, Mater. Sci. Semicond. Proc. 107 (2020) 104834, J. Ind. Eng. Chem. 76 (2019) 476, etc. In the current state, there are more typographical errors and the language should be improved. Therefore, the authors are advised to recheck the whole manuscript for improving the language and structure carefully.

Author Response

Response to Reviewer 1 Comments

Comments to the Author:

In this manuscript, a novel type-II heterojunction photocatalysts, MgSn(OH)6/SnO2, were successfully constructed through a facile one-pot in-situ hydrothermal method. The obtained products are thoroughly characterized by XRD, FT-IR, HRTEM, optical, and catalytic properties. The photocatalytic performance was examined for the degradation of a   tetracycline solution under UV light irradiation. The manuscript is well organized and contains interesting findings. However, I recommended a major revision of the article from its present form before it can be published in nanomaterials. The main concerns are listed below.

Technical Points

Point 1: The authors should explain the novelty of the present report in a scientific manner?

Response 1: Thank you very much for your kind suggestions! Construction of heterojunction was an effective method to accelerate carrier transfer and reduce the separation of photogenerated carriers spatially. A high-quality heterojunction with a good lattice match was highly desired. Herein, in this work, the heterojunction with high photocatalytic performance was fabricated by co-sharing of Sn atoms between the formed coherent or semi-coherent interface in atomic level. Thank you very much again!

Point 2: The authors should provide the schematic representation for the experimental procedure.

Response 2: Thank you very much for your kind suggestions. We have revised all the relevant part in the manuscript and highlighted, including all used reactants with their purity, the preparation of samples for FT-IR analysis, TOC measurement, the preparation of working electrode, calculations of the efficiency of the photodegradation of tetracycline and the methods of the performed recyclability tests.  Thank you very much again!

Point 3: The authors should include the line spectra and miller indices in the XRD pattern.

Response 3: Thank you very much for your kind notification and suggestions! We have added the miller indices of several strongest peaks in the XRD patterns and highlighted. Thank you very much again!

Figure 1 XRD patterns of SnO2(SO), MgSn(OH)6(MSOH), and MSOH-SO-3 heterojunction

Point 4: What is the interaction between MgSn(OH)6 and SnO2, which makes it stable? chemical bond? physical force? how to confirm that heterojunction really form by direct proof.? why is it not just a composite? although their band matches.

Response 4: Thank you very much for your kind comments and questions! The heterojunction between MgSn(OH)6(MSOH) and SnO2(SO) was designed and fabricated by co-sharing of Sn atoms in atomic level. We applied the mechanical mixing composites (MSOH and SO) to evaluate the photodegradation performance of the TC solution. As shown in Figure 2, when the mixture was used as the photocatalyst, the degradation rate of TC solution was only 40% within 60 min, which was much smaller than that of heterojunctions prepared via hydrothermal treatment. Therefore, we believed that a close contact interface was built by forming chemical bonds between MSOH and SO. Thank you very much again!

Figure 2 Photodegradation of TC solution with the synthesized catalysts (MSOH, SO, heterojunctions and mixture of MSOH and SO) under UV irradiation.

Point 5: How to exclude the sensitization mechanism of tetracycline for the present work?

Response 5: Thank you very much for your kind comments and suggestions. As shown in Figure 3, the energy levels of tetracycline molecule was added to better comprehend the photocatalytic mechanism (Analytica Chimica Acta 2019, 1063, 144-151.). The excited electrons on the CB of MgSn(OH)6 and LOMO level of tetracycline molecule tended to flow down to the CB of SnO2. However, the accumulated electrons in the CB position of SnO2 were more positive than O2/·O2, thus ·O2 would theoretically not participate in the photocatalytic reaction. In contrast, the remaining holes on the VB of SnO2 moved to the VB of MgSn(OH)6 and the HOMO level of tetracycline molecule. Thus, we speculated that the photodegradation of TC solution was mainly dominated by the oxidation of photoinduced h+ directly. The proposed mechanism was in good agreement with the trapping experiments of reactive species. Thank you very much again!

Figure 3 Schematic of the carrier transfer in a type-II heterojunction between MgSn(OH)6 and SnO2

Point 6: TOC is needed to sure the mineralization ratio.

Response 6: Thank you very much for your kind suggestions. Total dissolved organic carbon was determined via a TOC Analyzer (SHIMADZU, TOC-L CPB). The TOC removal rate of mineralizing TC molecule could reach 35% within 60 min. We have added the part in our paper and highlighted. Thank you very much again!

Figure 4 The TOC degradation of TC solution under UV-light irradiation.(TOC0: initial TOC value; TOCt: TOC value at different photodegradation time t)

Point 7: Photoluminescence spectra are needed to explore the mechanism.

Response 7: Thank you very much for your kind suggestions. The  reactive species were detected using fluorescence spectroscopy. As shown in Figure 5, the increase of fluorescence intensity was enhanced with prolonged irradiation time. Also, based on the proposed photocatalytic mechanism(Figure 3), the accumulated holes h+ in MgSn(OH)6 could capture OH- to generate ·OH thermodynamically. However, there was no large decrease in the degradation of the TC solution in the presence of 2-propanol (IPA) according to our reactive species trapping experiments. These results indicated that the produced  would not be favorable for the oxidation of TC (J. Phys. Chem. C. 116 (2012) 5764.).Thank you very much again!

Figure 5 Hydroxyl radical .OH detecting photoluminescence (PL) spectra of MSOH-SO-3 in terephthalic acid (TA) solution under UV light irradiation.

Point 8: The authors should check the loading effect of the catalyst for the degradation of dye.

Response 8: Thank you very much for your kind suggestions. The dark adsorption–desorption equilibrium of the as-prepared samples in tetracycline solution was established within 120 min. Thank you very much again!

Figure 6 The dark adsorption of MSOH, SO and MSOH-SO-3 in tetracycline solution(TC).

Point 9: The latest, important publications are suggested, such as, Mater. Sci. Semicond. Proc. 107 (2020) 104834, J. Ind. Eng. Chem. 76 (2019) 476, etc.

Response 9: Thank you very much for your kind suggestions. We have cited the paper “J. Ind. Eng. Chem. 76 (2019) 476” in our manuscript. However, the paper “Mater. Sci. Semicond. Proc. 107 (2020) 104834” was not found. Could you provide us with DOI of the paper? Thank you very much again!

Point 10: In the current state, there are more typographical errors and the language should be improved. Therefore, the authors are advised to recheck the whole manuscript for improving the language and structure carefully.

Response 10: Thank you very much for your comments and suggestions patiently. The whole manuscript was carefully checked. Thank you very much again!

Reviewer 2 Report

MS No: 

nanomaterials-655253-peer-review-v1

Title:

In situ construction of a MgSn(OH)6 perovskite/SnO2 type-II heterojunction for photodegradation of tetracycline

Authors:     

Yuanyuan Li, Xiaofang Tian, Yaoqiong Wang, Qimei Yang, Yue Diao, Bin Zhang and Dingfeng Yang

Evaluation: The present manuscript deals with the synthesis and characterization of a series of MgSn(OH)6 perovskite/SnO2 type-II heterojunctions. Their photocatalytic activity is tested for tetracycline degradation under UV irradiation. It is concluded that the formation of the heterojunction promotes photocatalytic activity due to the effective separation of photoinduced carriers. Photoelectrochemical measurements were carried out in order to investigate the photocatalytic mechanism and a photocatalytic mechanism is proposed.

In general, it is a well written and easy to follow manuscript. However, the use of UV responsive materials rather than visible light responsive, lowers the novelty of the present paper.

Overall, I believe that some major points should be addressed before publication on Nanomaterials

Below are some specific comments:

Authors used a 175 W high pressure mercury lamp with a primary wavelength of 365 nm. Authors should perform a chemical actinometry in order to find out the incident radiation intensity entering their photoreactor. Tetracycline concentration was 20 mg/L, too far from the real one’s concentrations in water matrices. Authors present EIS measurements but the corresponding experimental part is missing (film fabrication, electrolyte and electrodes used). Such information should be added. Moreover, I would advise the authors to assess the stability of the as prepared photocatalyst through chronoamperometry measurements. I would advise the authors to add information about the way they performed the recyclability tests. I would advise the authors to add information about the specific surface area of the synthesized materials. Conclusions must be rewritten.

Author Response

Response to Reviewer 1 Comments

Comments to the Author:

The present manuscript deals with the synthesis and characterization of a series of MgSn(OH)6 perovskite/SnO2 type-II heterojunctions. Their photocatalytic activity is tested for tetracycline degradation under UV irradiation. It is concluded that the formation of the heterojunction promotes photocatalytic activity due to the effective separation of photoinduced carriers. Photoelectrochemical measurements were carried out in order to investigate the photocatalytic mechanism and a photocatalytic mechanism is proposed. In general, it is a well written and easy to follow manuscript. However, the use of UV responsive materials rather than visible light responsive, lowers the novelty of the present paper. Overall, I believe that some major points should be addressed before publication on Nanomaterials.

Technical Points

Point 1: Authors used a 175 W high pressure mercury lamp with a primary wavelength of 365 nm. Authors should perform a chemical actinometry in order to find out the incident radiation intensity entering their photoreactor.

Response 1: Thank you very much for you kind suggestions! The lamp was positioned 9 cm above the reaction vessel and the light intensity on the surface of the suspension was measured approximately 11 mW/cm2. We have added the part in the section of Materials and Methods and highlighted. Thank you very much again!

Point 2: Tetracycline concentration was 20 mg/L, too far from the real one’s concentrations in water matrices.

Response 2: Thank you very much for your kind comments. We are totally agree with your concern. According to the previous reports, 20~50 mg photocatalysts were put into the tetracycline aqueous solution with concentration of 10~25 mg/L to conduct the photocatalytic performance(Inorg. Chem. Front., 2018, 5, 3148-3154; J. Colloid Interface Sci. 2018,519, 255-262; Appl. Catal. B Environ. 2018, 229, 96-104; ACS Sus. Chem. Eng. 2018, 6, 6941-6949.). Based on this, in our work, the concentration of tetracycline solution was set to be 20 mg/L and the added amount of photocatalyst was 30 mg. Thank you very much again!

Point 3: Authors present EIS measurements but the corresponding experimental part is missing (film fabrication, electrolyte and electrodes used). Such information should be added. Moreover, I would advise the authors to assess the stability of the as prepared photocatalyst through chronoamperometry measurements.

Response 3: Thank you very much for your kind notification and suggestions! For a typical preparation of working electrode, 10 mg of catalyst samples were dispersed in 500  of water/ethanol (240/250) mixed solvent containing 10  of 5wt% Nafion and sonicated for 30 min. Then, a certain amount of the catalyst ink was loaded onto a ITO electrode. Electrochemical impedance spectroscopy (EIS) was carried out using an alternating voltage with an amplitude of 5 mV over the frequency range of 105 Hz to 0.1 Hz and an open circuit voltage in 0.5 M Na2SO4. We have added the description in the paper and highlighted. Additionally, the time dependent current density for MSOH-SO-3 heterojunction was presented in Figure 1. It was clearly observed that the current density only showed slight degradation within 4 hours, indicating that sample MSOH-SO-3 exhibited superior electrochemical stability and durability. Thank you very much again!

Figure 1. The time-dependent current density curves for MSOH-SO-3 catalysts under UV light irradiation.

Point 4: I would advise the authors to add information about the way they performed the recyclability tests.

Response 4: Thank you very much for your kind suggestions! For each photocatalytic reaction, the catalysts were separated from the solution and dried at 60℃. Considering the small loss of the catalysts in the recycling process, several batches of repeated experiments for each cycle were performed. Then, the catalysts were collected and mixed to maintain the weight of 30 mg for each test. We have added the description in the paper and highlighted. Thank you very much again!

Point 5: I would advise the authors to add information about the specific surface area of the synthesized materials.

Response 5: Thank you very much for your kind comments and suggestions. We have measured the BET specific surface areas of the as-synthesized samples, heterojunction MSOH-SO-3, SnO2 (SO) and MgSn(OH)6 (MSOH). As presented in Figure 2, the results showed that the calculated specific surface areas of sample MSOH-SO-3 and SO were 114.2 m2/g and 81.7 m2/g, respectively. However, the obtained specific surface area of MgSn(OH)6 (MSOH) was too small to ignore. We have added the part in our paper and highlighted. Thank you very much again!

Figure 2. The specific surface area determination performed by BET measurement with N2 adsorption-desorption. (a) Heterojunction(MSOH-SO-3) (b) SnO2(SO).

Point 6: Conclusions must be rewritten.

Response 6: Thank you very much for your kind suggestions. We have rewritten our conclusions. Thank you very much again!

Reviewer 3 Report

The manuscript reports the fabrication of MgSn(OH)6/SnO2 heterojunctions, using a hydrothermal method, and characterization by powder X-ray diffraction, Fourier transform infrared spectroscopy, transmission electron microscopy, and ultraviolet-visible diffuse reflectance spectroscopy.  The goal is to evaluate the photocatalytic degradation of a tetracycline solution under ultraviolet irradiation.  The manuscript claims that the heterojunctions reported here exhibit superior photocatalytic activity compared with both parent compounds, due to the formation of a type-II junction photocatalytic system, more efficient in the transfer and separation of photogenerated electrons and holes. Additionally, it is claimed that the holes were the dominant active species, whereas hydroxyl radicals and superoxide ions had minor effects in the photodegradation of tetracycline.

Novelty.  The use of MgSn(OH)6 as photocatalyst for the photodegradation of pollutants is relatively recent.  The use of MgSn(OH)6 together with SnO2 to make heterojunctions is, to my knowledge, novel.  The hypothesis that making MgSn(OH)6/SnO2 heterojunctions would lead to improved charge separation and more efficient photodegradation is interesting and worth exploring.

Relevance and impact.  The topic is of high scientific and practical interest as the present study has relevance in the environmental restoration, particularly for the pharmaceutical residues in wastewater.  The manuscript reports the fabrication & characterization as well as the testing of the material for the proof of concept.  Obtaining photocatalysts with improved performance is a topic of intense current research. 

Scientific soundness.  Overall, the approach has rigor, with multiple and very diverse methods of characterization, each revealing a peace of the puzzle.  The paper is well documented and logically written.  However, not enough attention has been given to details and some conclusions are not supported by data. 

The legend of Fig 1b) needs to be improved, to tell the FTIR spectrum of SO apart from MSOH.  But, looking more carefully at the plot, one notices that the FTIR transmission goes beyond 150%!  How is that explained?  Do we see here a lasing effect, with light amplification?

Second, Fig 8 displays a schematic that does not raise to the level of rigor expected from a scientific article.  It may be amusing to see the sun on the upper left and the stars on the right of a blue sky, but they are only distractors.  Even worse, the depicted sun is deceiving, as the electromagnetic radiation used has the key emission in the UV at 365 nm, as stated on line 124 of page 3.  

Although very important for understanding the photodegradation mechanism Fig 8 lacks key information and, worse, is misleading, while the caption is not self-explanatory.  For instance, one needs to read the text very carefully to guess which oval represents which component of the compound system.  In addition, the two ovals do not touch, although the text emphasize the presence of a junction, which requires a common interface.  Worse is that the key molecular orbitals of tetracycline (the HOMO and the LUMO) are not shown such that the claim that MgSn(OH)6 could directly oxidize it remains unsubstantiated.  We need to be sure that the charge transfer can take place before making such statements.

Similarly, the claim that O.2- would not participate in the photocatalytic reaction because the electrons in SnO2 lay at lower energies is not totally convincing, as the electron transfer to O.2- may take place from MgSn(OH)6, at the other side of the junction.

Under these circumstances, I think that the novelty, relevance and impact of the present manuscript are significant.  The methods are appropriate and revealing.  What clearly needs a revision is the discussion of the photodegradation mechanism and of Fig. 8. 

Author Response

Response to Reviewer 2 Comments

Comments to the Author:

The manuscript reports the fabrication of MgSn(OH)6/SnO2 heterojunctions, using a hydrothermal method, and characterization by powder X-ray diffraction, Fourier transform infrared spectroscopy, transmission electron microscopy, and ultraviolet-visible diffuse reflectance spectroscopy. The goal is to evaluate the photocatalytic degradation of a tetracycline solution under ultraviolet irradiation. The manuscript claims that the heterojunctions reported here exhibit superior photocatalytic activity compared with both parent compounds, due to the formation of a type-II junction photocatalytic system, more efficient in the transfer and separation of photogenerated electrons and holes. Additionally, it is claimed that the holes were the dominant active species, whereas hydroxyl radicals and superoxide ions had minor effects in the photodegradation of tetracycline.

Novelty. The use of MgSn(OH)6 as photocatalyst for the photodegradation of pollutants is relatively recent. The use of MgSn(OH)6 together with SnO2 to make heterojunctions is, to my knowledge, novel. The hypothesis that making MgSn(OH)6/SnO2 heterojunctions would lead to improved charge separation and more efficient photodegradation is interesting and worth exploring.

Relevance and impact. The topic is of high scientific and practical interest as the present study has relevance in the environmental restoration, particularly for the pharmaceutical residues in wastewater. The manuscript reports the fabrication & characterization as well as the testing of the material for the proof of concept. Obtaining photocatalysts with improved performance is a topic of intense current research.

Scientific soundness. Overall, the approach has rigor, with multiple and very diverse methods of characterization, each revealing a peace of the puzzle. The paper is well documented and logically written. However, not enough attention has been given to details and some conclusions are not supported by data.

Technical Points

Point 1: The legend of Fig 1b) needs to be improved, to tell the FTIR spectrum of SO apart from MSOH.  But, looking more carefully at the plot, one notices that the FTIR transmission goes beyond 150%!  How is that explained?  Do we see here a lasing effect, with light amplification?

Response 1: Thank you very much for your kind questions and notification! For comparing with the FT-IR datas of as-prepared heterojunction MSOH-SO-3, SnO2 (SO) and MgSn(OH)6 (MSOH) in one picture distinctly, we added the value of Y-axis of MSOH-SO-3 and MgSn(OH)6 artificially. We apologized for our mistakes. The new plot was presented in Figure 1. Thank you very much again!

Figure 1 FT-IR spectra of SnO2, MgSn(OH)6 and MSOH-SO-3 heterojunction

Point 2: Fig 8 displays a schematic that does not raise to the level of rigor expected from a scientific article. It may be amusing to see the sun on the upper left and the stars on the right of a blue sky, but they are only distractors. Even worse, the depicted sun is deceiving, as the electromagnetic radiation used has the key emission in the UV at 365 nm, as stated on line 124 of page 3. Although very important for understanding the photodegradation mechanism Fig 8 lacks key information and, worse, is misleading, while the caption is not self-explanatory. For instance, one needs to read the text very carefully to guess which oval represents which component of the compound system. In addition, the two ovals do not touch, although the text emphasize the presence of a junction, which requires a common interface. Worse is that the key molecular orbitals of tetracycline (the HOMO and the LUMO) are not shown such that the claim that MgSn(OH)6 could directly oxidize it remains unsubstantiated. We need to be sure that the charge transfer can take place before making such statements. Similarly, the claim that·O2- would not participate in the photocatalytic reaction because the electrons in SnO2 lay at lower energies is not totally convincing, as the electron transfer to·O2- may take place from MgSn(OH)6, at the other side of the junction. Under these circumstances, I think that the novelty, relevance and impact of the present manuscript are significant. The methods are appropriate and revealing. What clearly needs a revision is the discussion of the photodegradation mechanism and of Fig. 8.

Response 2: Thank you very much for very kind comments and suggestions! Based on your kind advice, we have made the corresponding revision of the photodegradation mechanism plot. As shown in Figure 2, the energy levels of tetracycline molecule was added to better comprehend the photocatalytic mechanism (Analytica Chimica Acta 2019, 1063, 144-151.). The excited electrons on the CB of MgSn(OH)6 and LOMO level of tetracycline molecule tended to flow down to the CB of SnO2 and the accumulated electrons in the CB position of SnO2 were more positive than O2/·O2, thus ·O2 would theoretically not participate in the photocatalytic reaction. In contrast, the remaining holes on the VB of SnO2 moved to the VB of MgSn(OH)6 and HOMO level of tetracycline molecule. Thus, the h+ could directly oxidize the TC solution to the corresponding degradation products. Thank you very much again!

Figure 2 Schematic of the carrier transfer in a type-II heterojunction between MgSn(OH)6 and SnO2

Reviewer 4 Report

Ms. Ref. No.: NANOMATERIALS-655253

Title: “In situ construction of a MgSn(OH)6 perovskite/SnO2 type-II heterojunction for photodegradation of tetracycline

Authors: Yuanyuan Li, Xiaofang Tian, Yaoqiong Wang, Qimei Yang, Yue Diao, Bin Zhang and Dingfeng Yang

The manuscript concerns the hydrothermal synthesis of novel type-II heterojunction MgSn(OH)6/SnO2 products. The synthesized materials were used as an effective photocatalysts in photocatalytic decomposition of tetracycline. In my opinion this manuscript is interesting, but it needs major revision before it will be consider for potential publication in Nanomaterials journal. My reservations and comments are given below.

1.    The title of the manuscript should be improved.

Abstract – Authors should supplement the information about: terms of the hydrothermal treatment, initial concentration of the tetracycline using in pfotocatalytic test, the efficiency of the photocatalytic degradation of tetracycline in the presence of synthesized materials. Phrase “crystal”, should be change by “crystalline”. Section 2 – all reactants which were used in the presented study should be supplemented, together with their purity. Section 2.1. – in this section should appear the abbreviation of all synthesized materials. Moreover, could You introduce information about the final pH of reaction mixture of SnO2, MgSn(OH)6 and MgSn(OH)6/SnO2. Could Authors explain, why did You present HRTEM, FTIR, EDS mapping and UV-Vis DRS results for selected samples? In my opinion You should consider addition the results for all synthesized samples to observe the differences between the synthesized materials. Did Authors can provide the criterion for selecting samples for physicochemical analysis? Authors should consider to present the EDS results for all synthesized materials in Table, to observed the content of individual elements? In my opinion – Authors should consider addition the BET and XPS analysis for synthesized samples to observe the differences between the products. 1b – description of axis OY should be improved. In my opinion Transmittance should be in (a.u.) without the units. Figure 1a – in my opinion the synthesized MSOH-SO samples are characterized with worst crystalline structure compare to MSOH and SO samples. In this situation, could You explain why in the presence of those materials the better decomposition of tetracycline was observed? I suppose that mentioned materials are characterized with higher BET surface area. In this situation, the tetracycline in the first will be adsorbed on the surface on the photocatalysts. This situation was observed on Fig. 4a, after adsorption/desorption equilibrium almost 40% of tetracycline was adsorbed on MSOH-SO-4 sample. Section 2.1. – please verified the amount of the used precursors because sometimes Authors give amount in “mmol”, another time in “g”. Section 2.1. – Did Authors realize the synthesis of MgSn(OH)6/SnO2 materials via hydrothermal method in different temperature conditions, and check how it effects on the physicochemical properties od synthesized products? Those research will increase the scientific value of the manuscript. Write in the experimental part the preparation of sample for FTIR analysis. Section 2.3. – could You introduce information about the pH in which the photocatalysis test was realized, intervals in which the samples were taken and how long the photocatalytic process was carried out. Could Authors give information how did they calculate the efficiency of the decomposition of tetracycline?, Moreover, Authors should give the equation of the calibration curve of tetracycline. In my opinion in Section 2 Authors should supplement the information concern kinetic study of the photocatalytic study. The results of decomposition of tetracycline should be accompanied with the statistical treatment. In section 2.3. write in more details how did You realized the trapping experiment.

18.  In my opinion all abbreviation which appear in the manuscript, should be explain when they appear for the first time.

Description of samples, presented in legend in all figures, should be presented in the same style. To increase the scientific value of the manuscript, all presented results should compare with the results described in previous publications in this range. 2b – present the EDS mapping of selected sample, not spectrum of this sample. Please verified description of this figure. 4b – legend was omitted. 4d – Did Author explain why the curves concern 1 and 2 cycles start at C/C0=0.7 not at 1? Did Authors identify the degradation products by GC-MS or LC-MS? I recommend to check the degradation products after photocatalytic tests. I recommend to perform tests to determine the degree of mineralization of tetracycline through TOC analysis. Line 227 – is: “Figure 6a”, should be: “Figure 6”. 8 – could You specified the materials in ovoid in yellow and violet colour? I recommend Authors to check the whole manuscript, because there are a lot of editorial mistakes (typos, no spaces, etc.). English should be improved. There are several badly constructed sentences with grammatical errors. English should be checked by a native speaker.

Author Response

Response to Reviewer 3 Comments

Comments to the Author:

The manuscript concerns the hydrothermal synthesis of novel type-II heterojunction MgSn(OH)6/SnO2 products. The synthesized materials were used as an effective photocatalysts in photocatalytic decomposition of tetracycline. In my opinion this manuscript is interesting, but it needs major revision before it will be consider for potential publication in Nanomaterials journal. My reservations and comments are given below.

Technical Points

Point 1: The title of the manuscript should be improved.

Response 1: Thank you very much for your kind suggestions! The title of the manuscript was changed to “In situ construction of a MgSn(OH)6 perovskite/SnO2 type-II heterojunction: a highly efficient photocatalyst towards photodegradation of tetracycline”. Thank you very much again!

Point 2: Abstract-Authors should supplement the information about: terms of the hydrothermal treatment, initial concentration of the tetracycline using in photocatalytic test, the efficiency of the photocatalytic degradation of tetracycline in the presence of synthesized materials.

Response 1: Thank you very much for your kind suggestions! We have revised the relevant part in the Abstract section and highlighted. Thank you very much again!

Point 3: Phrase “crystal”, should be change by “crystalline”.

Response 3: Thank you very much for your kind suggestions! We have replaced “crystal” with “crystalline” in the maintext and highlighted. Thank you very much again!

Point 4: Section 2-all reactants which were used in the presented study should be supplemented, together with their purity.

Response 4: Thank you very much for your kind suggestions! MgCl2·6H2O and SnCl4·5H2O were purchased from the Aladdin Co.Ltd with purity AR. All the chemicals were directly utilized without further purification. We have added the part in our paper and highlighted. Thank you very much again!

Point 5: Section 2.1. – in this section should appear the abbreviation of all synthesized materials. Moreover, could You introduce information about the final pH of reaction mixture of SnO2, MgSn(OH)6 and MgSn(OH)6/SnO2.

Response 5: Thank you very much for your kind notifications. The abbreviation of SnO2 and MgSn(OH)6 was SO and MSOH, respectively. We have added the part in section 2.1 and highlighted. We speculated that the final pH of the reaction mixture of the as-synthesized photocatalysts were alkaline due to the excessive NaOH as the reactant. Thank you very much again!

Point 6: Could Authors explain, why did You present HRTEM, FTIR, EDS mapping and UV-Vis DRS results for selected samples? In my opinion You should consider addition the results for all synthesized samples to observe the differences between the synthesized materials. Did Authors can provide the criterion for selecting samples for physicochemical analysis? Authors should consider to present the EDS results for all synthesized materials in Table, to observe the content of individual elements?

Response 6: Thank you very much for you kind questions! The heterojunctions prepared with different ratios exhibited superior photocatalytic activity compared with the parent MgSn(OH)6 and SnO2 compounds. Among the investigated heterojunctions, MSOH-SO-3 showed the maximum photodegradation efficiency conversion of 91% within 60 min under ultraviolet irradiation. Therefore, we used the sample MSOH-SO-3 for physicochemical analysis to study the mechanism of enhanced photodegradation efficiency. 

Figure 1 elemental mapping of heterojunction MSOH-SO-3

Figure 1 showed the elemental composition of heterojunction MSOH-SO-3. Clearly, it showed the presence of all elements, i.e. Mg, Sn and O atoms. Thank you very much again!

Point 7: In my opinion – Authors should consider addition the BET and XPS analysis for synthesized samples to observe the differences between the products.

Response7: Thank you very much for your kind comments and suggestions. We have measured the BET specific surface areas of the as-synthesized samples, heterojunction MSOH-SO-3, SnO2 (SO) and MgSn(OH)6 (MSOH). As presented in Figure 2, the results showed that the calculated specific surface areas of sample MSOH-SO-3 and SO were 114.2 m2/g and 81.7 m2/g, respectively. However, the obtained specific surface area of MgSn(OH)6 (MSOH) was too small to ignore. We have added the part in our paper and highlighted. Thank you very much again!

Figure 2. The specific surface area determination performed by BET measurement with N2 adsorption-desorption. (a) Heterojunction(MSOH-SO-3) (b) SnO2(SO).

Point 8: 1b – description of axis Y should be improved. In my opinion Transmittance should be in (a.u.) without the units.

Response 8: Thank you very much for your kind notification! The new plot was presented in Figure 1. Thank you very much again!

Figure 3 FT-IR spectra of SnO2, MgSn(OH)6 and MSOH-SO-3 heterojunction

Point 9: Figure 1a – in my opinion the synthesized MSOH-SO samples are characterized with worst crystalline structure compare to MSOH and SO samples. In this situation, could You explain why in the presence of those materials the better decomposition of tetracycline was observed? I suppose that mentioned materials are characterized with higher BET surface area. In this situation, the tetracycline in the first will be adsorbed on the surface on the photocatalysts. This situation was observed on Fig. 4a, after adsorption/desorption equilibrium almost 40% of tetracycline was adsorbed on MSOH-SO-4 sample.

Response 9: Thank you very much for you kind questions and comments! In contrast to the parent compound MgSn(OH)6 and SnO2, heterojunction MSOH-SO-3 embraced larger specific surface areas. Therefore, we speculated that the enhanced photocatalytic performance could be ascribed to the large specific surface area and the formation of a type-II heterojunction that was beneficial for the efficient transfer and separation of photogenerated electrons and holes. Thank you very much again!

Point 10: Section 2.1. – please verified the amount of the used precursors because sometimes Authors give amount in “mmol”, another time in “g”.

Response 10: Thank you very much for your kind suggestions! We kept our unit the same in section 2.1. Thank you very much again!

Point 11: Section 2.1. – Did Authors realize the synthesis of MgSn(OH)6/SnO2 materials via hydrothermal method in different temperature conditions, and check how it effects on the physicochemical properties of synthesized products? Those research will increase the scientific value of the manuscript.

Response 11: Thank you very much for your kind comments! The temperature effect in hydrothermal treatment is very interesting in photocatalysis. We will conduct the research next. Thank you very much again for your kind suggestions!

Point 12: Write in the experimental part the preparation of sample for FTIR analysis.

Response 12: Thank you very much for your kind suggestions! The photocatalysts were fixed within a pressed KBr pellet. For instance, 1 mg heterojunction MSOH-SO-3 within 100 mg KBr were pressed at 15 MPa for 15 min, forming 13 mm pellets. Thus, Fourier transform infrared (FT-IR) spectra were collected using a Nicolet 360 spectrometer with a 2 cm–1 resolution in the range of 500–4000 cm−1. We have added the part in our paper and highlighted. Thank you very much again!

Point 13: Section 2.3. – could you introduce information about the pH in which the photocatalysis test was realized, intervals in which the samples were taken and how long the photocatalytic process was carried out.

Response 13: Thank you very much for your kind questions! The value of pH was tested to be 5.5 before the irradiation. As for the steps of photocatalysis, 5 mL aliquots of the suspension were taken at given time intervals (20 min) and separated by centrifugation. The process of photocatalytic performance was carried out within 60 min for each photocatatlytic experiment. Thank you very much again!

Point 14: Could Authors give information how did they calculate the efficiency of the decomposition of tetracycline? Moreover, Authors should give the equation of the calibration curve of tetracycline. In my opinion in Section 2 Authors should supplement the information concern kinetic study of the photocatalytic study. The results of decomposition of tetracycline should be accompanied with the statistical treatment.

Response14: Thank you very much for your kind questions and suggestions! The concentration of the TC solution was determined by UV-Vis spectrometry at 355 nm. The degradation rate could be calculated by the following formula:

                                        Photodegadation (%) = 1-C/C0

Where C0 is the absorbance of the initial solution and C is the absorbance of solution at a given time after the photocatalytic reaction.

The standard deviations of kinetic rate constants for heterojunction MSOH-SO-3 to degrade TC solutions of the four cycling performance was 0.00299. We have added the part in our paper and highlighted. Thank you very much again!

Point 15: In section 2.3, write in more details how did You realized the trapping experiment.

Response 15: Thank you very much for your kind suggestions! Trapping experiments of the active species were carried out using 30 mg of MSOH-SO-3 and 100 mL of TC solution (20 mg/L). The reactive intermediate participating in the degradation process was identified by using different sacrificial agents. For instance, 10 mL of 2-propanol (IPA), 0.1 mmol disodium ethylenediaminetetraacetic acid (EDTA), and 0.1 mmol 2,2,6,6-tetramethylpiperidine-1-oxyl (TEMPO) were added sequentially to probe ·OH, h+, and ·O2 radicals, respectively. We have added the part in our paper and highlighted. Thank you very much again!

Point 16: In my opinion all abbreviation which appear in the manuscript, should be explain when they appear for the first time.

Response 16: Thank you very much for your kind suggestions! We have revised all the abbreviation that appeared for the first time in this paper. Thank you very much again!

Point 17: Description of samples, presented in legend in all figures, should be presented in the same style.

Response 17: Thank you very much for your kind suggestions! We have revised the relevant part in the manuscript. Thank you very much again!

Point 18: To increase the scientific value of the manuscript, all presented results should compare with the results described in previous publications in this range.

Response 18: Thank you very much for your kind suggestions! We applied the commercial TiO2 (Anatase Phase) to evaluate the photodegradation performance of the TC solution. As shown in Figure 4, when TiO2 was used as the photocatalyst, the degradation rate of TC solution was 42% within 60 min, while the rate of heterojunction MSOH-SO-3 could reach to be 91%. Therefore, sample MSOH-SO-3 exhibited excellent performance for the degradation of TC solution compared with commercial Anatase phase TiO2. Thank you very much again!

Figure 4 Photocatalytic degradation of TC solution with heterojunction MSOH-SO-3 and commercial TiO2 (Anatase Phase) under UV light irradiation.

Point 19: 2b – present the EDS mapping of selected sample, not spectrum of this sample. Please verified description of this figure. 4b – legend was omitted.

Response 19: Thank you very much for your kind notifications and suggestions! The elemental composition of heterojunction MSOH-SO-3 was shown in Figure 5. It showed the presence of all elements, i.e. Mg, Sn and O atoms. Thank you very much again!

Figure 5 elemental mapping of heterojunction MSOH-SO-3

We also added the legend of the fitted kinetic constant for photodegradation of TC solution in Figure 4b. Thank you very much again!

Point 20: 4d – Did Author explain why the curves concern 1 and 2 cycles start at C/C0=0.7 not at 1?

Response 20: Thank you very much for your kind questions! For each cycle performance, the photocatalyst and TC solution were fully stirred in the dark to establish the adsorption–desorption equilibrium. The absorbance was different after each dark adsorption experiment before the photochemical reaction was carried out. Thank you very much again!

Point 21:  Did Authors identify the degradation products by GC-MS or LC-MS? I recommend to check the degradation products after photocatalytic tests.

Response 21: Thank you very much for your kind suggestion! The intermediates during the photocatalytic degradation could be determined by the GC-MS or HPLC-MS. As indicated from previous reports (Chem. Eng. J. 2018, 333, 423-433.), the photogenerated reactive species could degrade TC molecule to low molecular weight organic compounds, such as 3-hexanone and alkanolamine, through dehydroxylation, deamination and ring opening reaction. Further, some intermediates were mineralized into CO2 and H2O. In this work, we applied the UV-vis spectrophotometer and TOC analysis to evaluate the photodegradation efficiency. Thank you very much again!

Point 22: I recommend to perform tests to determine the degree of mineralization of tetracycline through TOC analysis.

Response 22: Thank you very much for your kind suggestions! Total dissolved organic carbon was determined via a TOC Analyzer (SHIMADZU, TOC-L CPB). The TOC removal rate of mineralizing TC molecule could reach 35% within 60 min. We have added the part in our paper and highlighted. Thank you very much again!

Figure 6 The TOC degradation of TC solution under different UV-light irradiation.(TOC0: initial TOC value; TOCt: TOC value at photodegradation time t)

Point 23: Line 227 – is: “Figure 6a”, should be: “Figure 6”. Could you specified the materials in ovoid in yellow and violet color?

Response 23: Thank you very much for your kind notifications! We have revised the relevant part in the paper. Thank you very much again!

Point 24: I recommend Authors to check the whole manuscript, because there are a lot of editorial mistakes (typos, no spaces, etc.). English should be improved. There are several badly constructed sentences with grammatical errors. English should be checked by a native speaker.

Response 24: Thank you very much for your comments and suggestions patiently. The whole manuscript was carefully checked. Thank you very much again!

Round 2

Reviewer 1 Report

The authors are responded in a proper manner for the comments and the manuscript is well organized in the revised version.  Hence, it can be acceptable in the present form.

Author Response

Thanks very much for your kind work and consideration on publication of our paper. We would like to express our great appreciation to the reviewer.

Reviewer 2 Report

Accept

Author Response

(The authors gave the same response as above.)

Reviewer 4 Report

1. Abstract – Authors should supplement the information about temperature and time of the hydrothermal synthesis of MgSn(OH)6/SnO2 materials. Moreover, Authors could specify the ratio of MgSn(OH)6 to SnO2 (molar or weight) in which materials were synthesized. 2. All reactants (include: NaOH, tetracycline, ethanol, etc.) which were used in the experiments should be supplemented, together with their purity. 3. In section 2.1 Authors should introduce information about the final value of pH of reaction mixture of SnO2, MgSn(OH)6 and MgSn(OH)6/SnO2. 4. Authors should add the results of FTIR, EDS, UV-Vis DRS analysis for all synthesised samples to observe the differences between the MgSn(OH)6/SnO2 materials. Those results will increase the scientific value of the manuscript, because then we can observe how changes in ratio of components affected on their physicochemical properties. 5. In my opinion EDX analysis was used to determine the surface composition of the synthesized materials. Moreover, Authors should add the EDS results for all synthesized materials in Table, to observed how the content of individual elements change after different ratio of components which build the MgSn(OH)6/SnO2 materials? 6. Fig. 1b – description of axis OY should be change. In my opinion Transmittance should be in (a.u.) without the units. 7. Write in the experimental part the details about BET analysis. 8. Section 2.3. – Authors should introduce in the manuscript the information about the pH in which the photocatalysis test was realized as well as intervals in which the samples were taken and how long the photocatalytic process was carried out. Moreover, Authors should also give the equation of the calibration curve of tetracycline. 9. In my opinion description concern TOC analysis presented in Section 2.2 should move to section 2.3. 10. Description of FTIR analysis – could You check that at 1175 cm-1 we can observed the signals of O-H bending vibrations? 11. Authors should present in the manuscript the adsorption/desorption isotherms for all synthesized samples to observe the differences between the products, because value of BET surface area is a crucial parameter which has significant influence of photocatalytic activity of synthesized materials. 12. Authors should add the information concern kinetic study of the photocatalytic study. 13. Description of XRD results should be supplement for all reflections visible on XRD patterns. 14. Description of samples presented in all figures, should be presented in the same style in whole manuscript. 15. Line 176 – is: “Ca”, should be: “Mg”. 16. Line 217 - Authors should indicate which rate constant values are characteristic for a given sample. 17. In my opinion results of TOC analysis is very low, which may prove that intermediate products of the photocatalytic process may be formed, so in my opinion it is worth doing the HPLC analysis. 18. Fig. 6 – in my opinion units of “Degradation rate (%)” should be presented from 0 to 100. 19. Authors should compare all presented results with the results described in previous publications in this range, to increase the scientific value of the manuscript. 20. Fig. 1a – XRD patterns of MSOH-SO-3 sample is characterized with low crystallinity. Probably in this situation the mentioned sample will be characterized with high BET surface area. In this situation, the tetracycline will be more effective adsorbed on the surface on the photocatalysts. 21. I recommend Authors to check the whole manuscript, because there are a lot of editorial mistakes (typos, no spaces, etc.).

Author Response

Response to Reviewer 4 Comments

Technical Points

Point 1: Abstract – Authors should supplement the information about temperature and time of the hydrothermal synthesis of MgSn(OH)6/SnO2 materials. Moreover, Authors could specify the ratio of MgSn(OH)6 to SnO2 (molar or weight) in which materials were synthesized.

Response 1: Thank you very much for your kind suggestions and comments! ” In this work, a novel type-II heterojunction photocatalyst, MgSn(OH)6/SnO2, was successfully prepared via a facile one-pot in situ hydrothermal method at 220 °C for 24 h. The prepared heterojunctions exhibited superior photocatalytic activity compared with the parent MgSn(OH)6 and SnO2 compounds. Among them, the obtained MgSn(OH)6/SnO2 heterojunction with MgCl2·6H2O:SnCl4·5H2O=4:5.2 (mmol) displayed the highest photocatalytic performance and the photodegradation efficiency conversion of 91% could be reached after 60 min under ultraviolet irradiation.” We have revised the relevant part in the abstract section and highlighted. Thank you very much again!

Point 2: All reactants (include: NaOH, tetracycline, ethanol, etc.) which were used in the experiments should be supplemented, together with their purity.

Response 2: Thank you very much for your kind notifications! The following materials were applied for this experiment: MgCl2·6H2O (Aladdin, 98%), SnCl4·5H2O(Aladdin, 99%), tetracycline(Damas-beta, 97%+), NaOH(Aladdin, 96%), ethyl alcohol(Aladdin, 99.7%), 2-propanol(IPA) (Aladdin, 99.7%), disodium ethylenediaminetetraacetic acid(EDTA)(Aladdin, 99.5%) and 2,2,6,6-tetramethylpiperidine-1-oxyl(TEMPO)( Aladdin, 98%). We have revised the relevant part and highlighted. All of the above chemicals were directly utilized without any further purification. Thank you very much again!

Point 3: In section 2.1, authors should introduce information about the final value of pH of reaction mixture of SnO2, MgSn(OH)6 and MgSn(OH)6/SnO2.

Response 3: Thank you very much for your kind questions! For the preparation of SnO2, the pH of reaction mixture solution was adjusted to ~7. However, to obtain pure MgSn(OH)6, the pH of the reaction solution was adjusted to ~11. For the synthesis of heterojunctions MgSn(OH)6/SnO2, the pH was adjusted to ~8. We have added the relevant part in our manuscript. Thank you very much again!  

Point 4: Authors should add the results of FTIR, EDS, UV-Vis DRS analysis for all synthesised samples to observe the differences between the MgSn(OH)6/SnO2 materials. Those results will increase the scientific value of the manuscript, because then we can observe how changes in ratio of components affected on their physicochemical properties.

Response 4: Thank you very much for your kind suggestions and comments! As indicated in the observed photodegradation experiments, the as-synthesized heterojunctions exhibited superior photocatalytic activity compared with the parent MgSn(OH)6 and SnO2 compounds. Specially, the heterojunction MSOH-SO-3(SnCl4·5H2O/MgCl2·6H2O = 5.2/4mmol) exhibited the maximum photodegradation efficiency conversion of 91% within 60 min under ultraviolet irradiation. Based on this, we only used the sample MSOH-SO-3 for physicochemical analysis to discover the mechanism of enhanced photodegradation performance. Thank you very much again!

Point 5: In my opinion EDX analysis was used to determine the surface composition of the synthesized materials. Moreover, Authors should add the EDS results for all synthesized materials in Table, to observed how the content of individual elements change after different ratio of components which build the MgSn(OH)6/SnO2 materials?

Response 5: Thank you very much for your kind suggestions! We only used sample MSOH-SO-3(exhibited the excellent photodegradation efficiency in these synthesized heterojunctions) to study the relevant photocatalytic properties. For sample MSOH-SO-3, we obtained the EDS results in two different regions and the results are listed in Table 1. It is observed that Mg, Sn and O are the main elements in the as-synthesized sample MSOH-SO-3. Thank you very much again!

Table 1. The EDS results of sample MSOH-SO-3 in two different regions.

Sample MSOH-SO-3

Mg

Sn

O

Atomic Fraction (%)

Region 1

12.14

36.70

51.16

Atomic Fraction (%)

Region 2

11.58

37.04

51.38

Point 6: Fig. 1b – description of axis OY should be change. In my opinion Transmittance should be in (a.u.) without the units.

Response 6: Thank you very much for your kind notification! We have revised the FT-IR spectrum of the obtained samples and highlighted. Thank you very much again!

Point 7: Write in the experimental part the details about BET analysis.

Response 7: Thank you very much for your kind suggestions! The BET specific surface areas were investigated by means of N2 adsorption-desorption at 77 K using a Quantachrome QuadraWin and the specific surface areas were determined according to the BET method in the relative pressure range p/p0=0.069~0.249. We have added relevant part in the experiment part and highlighted. Thank you very much again!

Point 8: Section 2.3. – Authors should introduce in the manuscript the information about the pH in which the photocatalysis test was realized as well as intervals in which the samples were taken and how long the photocatalytic process was carried out. Moreover, Authors should also give the equation of the calibration curve of tetracycline.

Response 8: Thank you very much for your kind questions! The value of pH was tested to be 5.5 before the irradiation. As for the steps of photocatalysis, 5 mL aliquots of the suspension were taken at given time intervals (20 min) and separated by centrifugation. The process of photocatalytic performance was carried out within 60 min. We have revised relevant part in section 2.3 and highlighted. As for the photodegradation of TC solution, the corresponding changes in the characteristic absorption curves are shown in Figure 1. Thank you very much again!

Figure 1 UV-visible absorption of TC as a function of irradiation time over the heterojunction MSOH-SO-3

Point 9: In my opinion description concern TOC analysis presented in Section 2.2 should move to section 2.3.

Response 9:  Thank you very much for your kind suggestions! We have moved the TOC analysis to section 2.3 and highlighted. Thank you very much again!

Point 10: Description of FTIR analysis – Could you check that at 1175 cm-1 we can observed the signals of O-H bending vibrations?

Response 10: Thank you very much for your kind questions! As described in previous work(Appl. Sur. Sci. 466 (2019) 515-524), peak at 1175 cm-1 was ascribed to the bending vibrations of O-H bonds. Thank you very much again!

Point 11: Authors should present in the manuscript the adsorption/desorption isotherms for all synthesized samples to observe the differences between the products, because value of BET surface area is a crucial parameter which has significant influence of photocatalytic activity of synthesized materials.

Response 11: Thank you very much for your kind comments! As suggested in our photocatalytic results, heterojunction MSOH-SO-3 exhibited the highest photodegradation efficiency among these prepared samples. Base on this, we chosen heterojunction MSOH-SO-3 as a representative to study the photocatalytic properties. Then, we measured the BET specific surface areas of samples, heterojunction MSOH-SO-3, SnO2 (SO) and MgSn(OH)6 (MSOH). As presented in Figure 2, the results showed that the calculated specific surface areas of sample MSOH-SO-3 and SO were 114.2 m2/g and 81.7 m2/g, respectively. However, the obtained specific surface area of MSOH was too small to ignore. As manifested of the photocatalytic results, there was approximately 32% of tetracycline absorbed on the MSOH-SO-3 after dark absorption-desorption equilibrium. In contrast, only 5% of tetracycline was absorbed on the MSOH or SO samples. The results are in good accordance with the observed BET values. Thank you very much again!

Figure 2. The specific surface area determination performed by BET measurement with N2 adsorption-desorption. (a) Heterojunction(MSOH-SO-3) (b) SO.

Point 12: Authors should add the information concern kinetic study of the photocatalytic study.

Response 12:  Thank you very much for your kind comments! The behavior of the photocatalytic reaction could be described by a pseudo-first order model:

                        -ln(C/C0) = kt   ,                                   

Where  and  are the TC concentrations in solution at time 0 and t, respectively; and  is the fitted kinetic rate constant. A linear relationship was observed between  and t based on the observed photocatalytic datas. The R2 of the linear fitting for MSOH, SO, MSOH-SO-1, MSOH-SO-2, MSOH-SO-3 and MSOH-SO-4 are 0.9746, 0.9979, 0.9136, 0.9099, 0.9524 and 0.9365, respectively. Thank you very much again!

Point 13: Description of XRD results should be supplement for all reflections visible on XRD patterns.

Response 13: Thank you very much for your kind notifications! We have indexed all the peaks in our observed XRD patterns. Thank you very much again!

Point 14: Description of samples presented in all figures, should be presented in the same style in whole manuscript.

Response 14: Thank you very much for your kind suggestions! We have revised the relevant descriptions in the manuscript. Thank you very much again!

Point 15: Line 176 – is: “Ca”, should be: “Mg”.

Response 15: Thank you very much for your kind notification! We have corrected it. Thank you very much again!

Point 16: Line 217 - Authors should indicate which rate constant values are characteristic for a given sample.

Response 16: Thank you very much for your kind suggestions patiently! The corresponding rate constants of MSOH-SO-1, MSOH-SO-2, MSOH-3 and MSOH-SO-4 were 0.018 min−1, 0.024 min−1, 0.030 min−1, and 0.028 min−1, respectively. We have revised relevant part and highlighted. Thank you very much again!

Point 17: In my opinion results of TOC analysis is very low, which may prove that intermediate products of the photocatalytic process may be formed, so in my opinion it is worth doing the HPLC analysis.

Response 17: Thank you very much for your kind questions and suggestions! The intermediates during the photocatalytic degradation could be determined by the GC-MS or HPLC-MS. As indicated from previous reports (Chem. Eng. J. 2018, 333, 423-433.), the photogenerated reactive species could degrade TC molecule to low molecular weight organic compounds, such as 3-hexanone and alkanolamine, through dehydroxylation, deamination and ring opening reaction. Further, some intermediates were mineralized into CO2 and H2O. Thank you very much again!

Point 18: Fig. 6 – in my opinion units of “Degradation rate (%)” should be presented from 0 to 100.

Response 18: Thank you very much for your kind notifications patiently! We have corrected it and highlighted. Thank you very much again!

Point 19: Authors should compare all presented results with the results described in previous publications in this range, to increase the scientific value of the manuscript.

Response 19: Thank you very much for your kind suggestions!  Among the as-prepared heterojunctions, MSOH-SO-3 exhibited the best photocatalytic ability and the photodegradation rate reached nearly 91% in 60 min, which was much larger than that of state-of-art UV light responsive ZnO(J. Haz. Mater. 324(2017) 329-339). We have added the relevant part and highlighted. Thank you very much again!

Point 20: Fig. 1a – XRD patterns of MSOH-SO-3 sample is characterized with low crystallinity. Probably in this situation the mentioned sample will be characterized with high BET surface area. In this situation, the tetracycline will be more effective adsorbed on the surface on the photocatalysts.

Response 20: Thank you very much for your kind comments! The observed specific surface areas of sample MSOH-SO-3 was 114.2 m2/g, which is much larger than that of parent compound MSOH(too small to ignore) and SO(81.7 m2/g). In general, samples with larger BET value are beneficial for the absorptions of molecules, leading to superior photocatalytic properties. As manifested of our photocatalytic results, there was approximately 32% of tetracycline absorbed on the MSOH-SO-3 after dark absorption-desorption equilibrium with 60 min. However, only 5% of tetracycline was absorbed on the MSOH or SO samples under the same condition. The results were in good accordance with the observed BET results. Thank you very much again!

Point 21: I recommend Authors to check the whole manuscript, because there are a lot of editorial mistakes (typos, no spaces, etc.).

Response 21: Thank you very much for your suggestions. The whole manuscript was carefully checked. Thank you very much again!
